# TOWARDS REALISTIC LONG-TAILED SEMI-SUPERVISED LEARNING IN AN OPEN WORLD

## ABSTRACT

Open-world long-tailed semi-supervised learning (OLSSL) has increasingly attracted attention. However, existing OLSSL algorithms generally assume that the distributions between known and novel categories are nearly identical. Against this backdrop, we construct a more *Realistic Open-world Long-tailed Semi-supervised Learning* (**ROLSSL**) setting where there is no premise on the distribution relationships between known and novel categories. Furthermore, even within the known categories, the number of labeled samples is significantly smaller than that of the unlabeled samples, as acquiring valid annotations is often prohibitively costly in the real world. Under the proposed ROLSSL setting, we propose a simple yet potentially effective solution called dual-stage post-hoc logit adjustments. The proposed approach revisits the logit adjustment strategy by considering the relationships among the frequency of samples, the total number of categories, and the overall size of data. Then, it estimates the distribution of unlabeled data for both known and novel categories to dynamically readjust the corresponding predictive probabilities, effectively mitigating category bias during the learning of known and novel classes with more selective utilization of imbalanced unlabeled data. Extensive experiments on datasets such as CIFAR100 and ImageNet100 have demonstrated performance improvements of up to 50.1%, validating the superiority of our proposed method and establishing a strong baseline for this task. For further researches, the experimental code will be open soon.

## 1 INTRODUCTION

In recent years, due to the prohibitive cost of labeling large amounts of data, many researchers have shifted their focus to semi-supervised learning (SSL). This learning paradigm aims to compensate for the lack of labeled data by leveraging the information from a large amount of unlabeled data. However, most existing semi-supervised learning methods Ahmed et al. (2020); Berthelot et al. (2019); Oliver et al. (2018); Chen et al. (2020) follow closed-set and class-balanced assumptions, which are unrealistic. The former assumption means that the labeled data, unlabeled data, and test data all contain samples of the same classes, but the unlabeled and test datasets often contain new classes that are not present in labeled dataset. For the latter assumption, it indicates that both labeled and unlabeled datasets are class-balanced, which conflicts the fact that the class distribution of real datasets is inevitably long-tailed. And long-tailed distribution causes a significant issue: there will be a large discrepancy in test accuracy between the head classes and the tail classes. To solve aforementioned problems, open-world semi-supervised learning (Open-world SSL) Cao et al. (2022); Sun & Li (2023); Mullappilly et al. (2024); Wang et al. (2023a) and long-tailed semi-supervised learning Kim et al. (2020a); Lai et al. (2022); Lee et al. (2021a); Wei & Gan (2023a); Wei et al. (2021b) have been proposed. Moreover, to simultaneously address open-world and long-tailed recognition problems, open-world long-tailed SSL (OLSSL) Bai et al. (2023); Zhang et al. (2023) is proposed to learn long-tailed and open-end data during training and test on a balanced test dataset containing samples from head, tail and open classes. Existing OLSSL methods follow a setting where the number of known classes is consistent with that of unknown classes in labeled data, which conflicts with real-world applications. The number of labeled data for known classes tends to be smaller than that of unlabeled data due to the expensive labeling cost. The realistic circumstance further increases the difficulty of recognizing known and novel classes.

Table 1: Relationship between our novel ROLSSL and other machine learning settings.

| Setting | Known classes | Novel classes | Data Distribution | S/N Consistency |
|---|---|---|---|---|
| Semi-supervised learning (SSL) | Classify | Not present | Balanced | Reject |
| Robust SSL | Classify | Reject | Balanced | Reject |
| Open-set recognition | Classify | Reject | Balanced | Reject |
| Open-set SSL | Classify | Not present | Balanced | Reject |
| Long-tailed SSL | Classify | Not present | Long-tailed | Reject |
| Generalized zero-shot learning | Classify | Discover | Balanced | Yes |
| Novel class discovery | Not present | Discover | Balanced | Yes |
| Open-world recognition | Classify | Discover | Balanced | Yes |
| Open-world SSL (OSS) | Classify | Discover | Balanced | Yes |
| Open-world long-tailed SSL (OLSSL) | Classify | Discover | Long-tailed | Yes |
| Realistic open-world long-tailed SSL | Classify | Discover | Long-tailed | No |

To simulate the real-world tasks, we propose a novel SSL setting named *Realistic Open-world Long-tailed Semi-supervised Learning* (**ROLSSL**). Unlike the OLSSL setting, the OLSSL setting, the training set employed for model training consists of a small amount of labeled data and abundant unlabeled data for known classes in ROLSSL setting, which greatly increases the difficulty of model recognition and classification of known classes. Furthermore, the class distributions of unlabeled data are categorized into three representative forms: *Consistent*, *Uniform*, and *Reversed*. The model not only needs to extract knowledge relevant to novel classes from a large amount of long-tailed unlabeled data to identify novel classes and assign instances to them, but also utilize the extracted information to assist in training on long-tailed labeled dataset with a small number of samples for classifying known classes. It indicates that higher requirements are placed on the recognition algorithm.

Due to the poor performance of the OLSSL algorithm under the ROLSSL setting and its tendency to degrade the recognition of novel classes as training progresses, the original PLA only maintains good performance in datasets with few classes (detailed in Section 4.3). To address the ROLSSL problem, we initially apply post-hoc logit adjustment (PLA) Menon et al. (2021) to the ROLSSL setting but find that the original PLA maintains good performance only in datasets with few classes, such as CIFAR-10 and SVHN. For datasets with more classes, it significantly reduces model performance (detailed in Ablation 4.4). Consequently, we revisit the design of PLA, incorporating sample frequency data, total class count, overall dataset size, and estimated sample frequency of unlabeled data to develop a dual-stage PLA (DPLA). By considering the relative context of current data, such as the total number of categories, the first-stage PLA adaptively modifies the relationship between the sample frequency of labeled data and the magnitude of logit adjustment, thereby encouraging a larger relative margin between the logits of rare and dominant labels in ROLSSL and preventing the degradation of novel class recognition during training. Furthermore, we aim to improve performance by making more effective use of unlabeled data. We apply the predicted sample frequency of the model to scale the logits for each class accordingly. In this process, we suppress the contribution of classes with higher frequency to the loss calculation while encouraging greater participation from classes with lower frequency. This approach, termed the second-stage PLA, helps the model achieve better recognition performance in the ROLSSL setting. Additionally, the first-stage PLA is utilized to adjust the generated pseudo-labels, further enhancing the model's performance.

The main contributions are summarized as follows:

- We propose a ROLSSL setting where the number of labeled data is much smaller than that of unlabeled data for known classes, and the distribution of labeled and unlabeled data mismatches, which better simulates the requirements of real-world applications.

- A novel strategy named dual-stage post-hoc logit adjustments (DPLA) is designed consisting of the first stage logit adjustment that integrates factors about sample frequency and the number of classes to better utilize labeled and unlabeled data and the second stage that guides model to suppress the participation to categories with more samples and encourage to make better use of less frequent categories.

- The detailed experimental results and ablation experiments demonstrate that the proposed ROLSSL setting is more difficult to be solved. And the DPLA strategy achieves excellent performance compared with previous advanced methods on six benchmark datasets.

The rest of this paper is organized as follows. Section 2 introduces some relevant work in the field of Long-tailed SSL and OLSSL. The proposed method is illustrated in Section 3 and experimental results are given in Section 4. Besides, conclusions are provided in Section 5.

## 2 RELATED WORK

**Long-tailed Semi-supervised Learning:** Long-tail semi-supervised learning (LTSSL) has garnered attention for its relevance in real-world applications. Various methods have been developed to tackle its challenges. Techniques such as DARP Kim et al. (2020b) and CReST Wei et al. (2021a) aim to correct biased pseudo-labels by aligning the distributions of labeled and unlabeled data. ABC Lee et al. (2021b) improves generalization by using an auxiliary classifier to adjust biases in predominant classes. CoSSL Fan et al. (2022) employs a mixup strategy Zhang et al. (2017) that focuses on minority classes to enhance performance. However, these methods often assume consistent distributions across labeled and unlabeled data, which may not hold true in practice. DASO Oh et al. (2022) offers a dynamic method that adjusts pseudo-labels using linear and semantic approaches based on observed class distributions. Despite its effectiveness, the issue of skewed class distributions still affects the accuracy of learned representations and pseudo-label reliability. ACR Wei & Gan (2023b) addresses this by introducing an Adaptive Consistency Regularizer that estimates and adjusts to the true class distribution of unlabeled data, facilitating more accurate pseudo-label refinement.

**Open-world Semi-supervised Learning (OSSL):** ORCA Cao et al. (2022) first proposed the OSSL task, recognizing that unlabeled test data may include classes not present in the labeled training set. It differs from novel class discovery Han et al. (2019; 2020); Zhao & Han (2021); Zhong et al. (2021) in that it does not assume that unlabeled data consists solely of new class samples. Recent advancements have aimed to enhance OSSL performance. OpenLDN Rizve et al. (2022) introduces a pairwise similarity loss to detect new classes, thereby converting the problem into a standard semi-supervised learning (SSL) task upon the discovery of new classes. OpenCon Sun & Li (2023) employs contrastive prototype learning to create a compact representation space that promotes tight clustering by aligning representations within the same predicted category. Further studies explore solutions for scenarios where known and unknown classes share a long-tail distribution (OLSSL). Bacon Bai et al. (2024) combines contrastive learning and pseudo-labeling to address imbalances in open-world recognition, while NCDLR Chuyu et al. (2023) uses a relaxed optimal transport problem to infer high-quality pseudo-labels for new classes, mitigating bias in learning known and new categories. Specifically, Realistic long-tailed open-world SSL (ROLSSL) differs from existing tasks by not assuming relationships between known and unknown category distributions, and by stipulating that labeled data in known categories is significantly less than their unlabeled counterparts.

## 3 METHOD

To avoid the imbalance between known and novel category data, which biases model learning towards dominant labels, we have revisited strategies based on label frequency for post-hoc logit adjustment and threshold tuning for pseudo label masks. Due to the complete failure of the original post-hoc logit adjustment in open-set long-tail recognition, which suppressed model performance compared to an unmodified learning process, the former reconsidered the relationship between label frequency, category count, and dataset size to encourage a larger relative margin between the logits of rare and dominant labels. The latter, on the other hand, relies on estimates of the categories to which unlabeled data belong, making targeted adjustments to the probabilities of pseudo labels predicted to belong to different categories to promote training of less numerous classes. This also involves masking pseudo labels of more numerous classes, allowing for the use of high-quality pseudo labels to mitigate their dominance in loss computation. We retain the fundamental open-world recognition framework, which leverages pairwise similarity loss to implicitly cluster unlabeled data into known and novel categories and uses entropy regularization to prevent a single category from dominating the batch.

### 3.1 PROBLEM FORMULATION

In the ROLSSL scenario, we consider three kinds of datasets: a labeled known-class dataset $\mathcal{D}_k^l$, an unlabeled known-class dataset $\mathcal{D}_k^u$, and an unlabeled novel-class dataset $\mathcal{D}_n^u$. The known-class

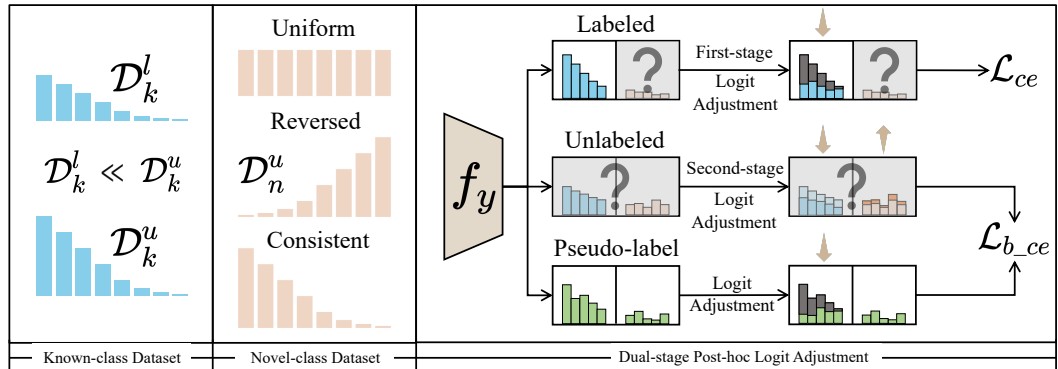

Figure 1: The overview of the ROLSSL setting and the Dual-stage Post-hoc Logit Adjustment method. On the left, the dataset composition within the ROLSSL framework is illustrated. On the right, the overall process of the Dual-stage Post-hoc Logit Adjustment is shown. In the first stage of logit adjustment, factors such as the number of classes, sample frequency, and overall dataset size are considered to encourage a larger relative margin between the logits of rare and dominant labels. In the second stage, the predicted class frequencies are used to adjust the logits for the unlabeled data, further guiding the model to focus on learning from predicted minority class samples and reducing the attention given to samples from the predicted majority classes.

dataset $\mathcal{D}_k^l$ consists of $m_k^l$ labeled samples $\{(x_i^l, y_i^l)\}_{i=1}^{m_k^l}$ and the unlabeled known-class dataset $\mathcal{D}_k^u$ consists of $m_k^u$ unlabeled samples $\{x_j^u\}_{j=1}^{m_k^u}$, where $x_i^l$ is a labeled instance with label $y_i^l = [c_k] = \{1, 2, \ldots, c_k\}$, and $x_j^u$ is an unlabeled instance from one of $c_k$ known classes, with $m_k^l \ll m_k^u$. Let $N_c$ represent the number of samples for class $c$ in the labeled known-class dataset, we have $N_1 \geq N_2 \geq \ldots \geq N_{c_k}$, and the imbalance ratio of the labeled known-class dataset can be denoted as $\gamma_k^l = \frac{N_1}{N_{c_k}}$. The unlabeled known-class dataset remains the same setting of the labeled known-class dataset and the number of samples for class $c$ is denoted as $H_c$ with imbalance ratio $\gamma_k^u$. For the unlabeled novel-class dataset, let $M_c$ represent the number of samples for class $c$ and the imbalance ratio $\gamma_n^u = \frac{max_c M_1}{min_c M_{c_k}}$ because there is no assumption about the distributions on the unlabeled novel-class dataset. Three kinds of representative distributions are considered, i.e., consistent, uniform, and reversed. Specifically, 1) for *Consistent* setting, we have $M_1 \geq M_2 \geq \ldots \geq M_{c_k}$ and $\gamma_k^l = \gamma_n^u$; 2) for *Uniform setting*, we have $M_1 = M_2 = \ldots = M_{c_k}$ and $\gamma_n^u = 1$; 3) for *Reversed* setting, we have $M_1 \leq M_2 \leq \ldots \leq M_{c_k}$ and $\gamma_k^l = 1/\gamma_n^u$. The unlabeled novel-class dataset $\mathcal{D}_n^u = \{x_j^u\}_{j=m_k^u+1}^{m_k^u+m_n^u}$ includes $m_n^u$ samples, each belonging to one of $c_n$ novel classes, where $c_n$ represents the total number of classes in $\mathcal{D}_n^u$, with $m_k^l + m_k^u \stackrel{.}{=} m_n^u$. Under the ROSSL framework, the combined unlabeled dataset $\mathcal{D}^u = \{\mathcal{D}_k^u, \mathcal{D}_n^u\}$ may contain samples from classes that are not present in the labeled dataset $\mathcal{D}^l = \{\mathcal{D}_k^l\}$, with the total class count $c_t$ in the open-world setting being $c_t = c_k + c_n$.

## 3.2 FOUNDATIONAL TECHNIQUES OF OSSL

To identify new classes, previous work employs a neural network Rizve et al. (2022), denoted as $f_\Psi$, for feature extraction. This network projects an input image $x \sim \mathbb{P}^\mathcal{Q}$, $\mathcal{Q} = m_k^u + m_k^l + m_n^u$ for unknown distribution $\mathbb{P}$, into a high-dimensional embedding space $\mathbb{Z}$ by transforming $x$ into its embedded representation $\mathbf{z} \in \mathbb{R}^d$. The set of all embeddings is represented by $\mathbb{Z}$, and $\mathbb{X}$ denote the sets of input images, respectively. The system recognizes both known and novel class samples by employing a classifier, $f_\Theta$, which maps embeddings $\mathbf{z}$ to a structured output space $f_\Theta : \mathbb{Z} \to \mathbb{R}^{c_k+c_n}$, where the first $c_k$ logits are associated with known classes and the remaining $c_n$ logits correspond to novel classes. The classifier outputs are converted into softmax probabilities $\hat{y} = \text{Softmax}(f_\Theta \circ f_\Psi(x))$ for further processing. The primary goal is to effectively discern novel classes while maintaining recognition of known classes, achieved through an objective function comprising three components: a pairwise similarity loss $\mathcal{L}_{pair}$, a cross-entropy loss $\mathcal{L}_{ce}$, and an entropy regularization term $\mathcal{L}_{reg}$. The pairwise similarity loss enhances class differentiation Hsu et al. (2017); Chang et al. (2017), the cross-entropy loss facilitates the classification of known and novel classes using true labels and generated pseudo-labels Han et al. (2019); Chapelle & Zien (2005), and

the entropy regularization prevents the model from settling on overly simplistic solutions Arazo et al. (2020):

$$\mathcal{L}_{ossl} = \mathcal{L}_{pair} + \mathcal{L}_{ce} + \mathcal{L}_{reg} \tag{1}$$

Following training with $\mathcal{L}_{ossl}$, samples corresponding to the $c_n$ logits in the output space are classified as belonging to novel classes. Eventually, novel class samples are added to the labeled set with the generated pseudo-labels, enabling the application of any standard closed-world semi-supervised learning (SSL) method, thereby leading to further performance improvements.

### 3.3 DPLA: Dual-Stage Post-hoc Logit Adjustment

We initially consider the post-hoc logit adjustment (PLA) for data where the frequency of samples corresponding to specific categories can be precisely obtained Menon et al. (2021); Tao et al. (2023); Wang et al. (2023b). Given a labeled known-class sample $x_i^l$, suppose we learn a neural network with logits $f_y(x_i^l) = w_y^\top \Phi(x_i^l)$, $f_y = f_\Theta \circ f_\Psi$. We predict the label $\arg\max_{y \in [c_k]} f_y(x_i^l)$. When trained with softmax cross-entropy, $p_y(x_i^l) \propto \exp(f_y(x_i^l))$ can be viewed as an approximation of $\mathbb{P}(y|x_i^l)$, predicting the label with the highest probability. In the first-stage post-hoc logit adjustment for known class, we propose a new prediction method for the known-class dataset with suitable $\tau_1 > 0$:

$$\text{argmax}_{y_i^l \in [c_k]} \exp\left(w_y^\top \Phi(x_i^l)\right) / \Omega_{y_i^l}^\tau = \text{argmax}_{y_i^l \in [c_k]} f_y(x_i^l) - \tau_1 \cdot \log \Omega_{y_i^l} \tag{2}$$

Specifically, $\Omega_{y_i^l}$ is a parameter synthesizing consideration of number of classes, the sample frequency and overall size of dataset, which can be defined as (detailed in Ablation 4.4):

$$\Omega_{y_i^l} = 10 \cdot (\lceil \mathcal{C}/\mathcal{C}_{base} \rceil) \cdot \sqrt{\mathcal{S}/\mathcal{S}_{base}} \cdot \mathcal{F}_{y_i^l} \tag{3}$$

where $\mathcal{C}$, $\mathcal{S}$ and $\mathcal{F}$ are the total number of classes, overall size of the estimated dataset, $\mathcal{C}_{base}$ and $\mathcal{S}_{base}$ are the basic discounting parameter for total number of classes and overall size of the dataset, and $\mathcal{F}_{y_i^l}$ represents the sample frequency of the category to which the corresponding label belongs. For $\tau \neq 1$, we apply temperature scaling to the logits, formulated as $\bar{p}_{y_i^l}(x_i^l) \propto \exp\left(\tau^{-1} \cdot w_{y_i^l}^\top \Phi(x_i^l)\right)$.

This adjustment is based on having access to the true probabilities $\mathbb{P}(y_i^l|x_i^l)$ and involves calibrating the probabilities through temperature scaling, commonly used in the context of distillation Hinton et al. (2015). These techniques help improve the model's generalization ability across different class distributions. For the second stage, given the unknown categories of the unlabeled data, we cannot perform post-hoc logit adjustments as with known-category data where sample frequencies are accessible Van Engelen & Hoos (2020). However, the imbalance in the unlabeled data necessitates corresponding logit adjustments. We propose a simple logit adjustment approach for the unlabeled data, which involves scaling the logits for each category based on the predictions of neural network $f$ on the categories for the unlabeled samples and the scaling weight $w_c$ for class $c$ can be defined as:

$$w_c = \sigma\left(\frac{\exp(-\pi_c^r)}{\exp(-\pi_{\max}^r)}\right) \cdot (\alpha - \beta) + \beta \tag{4}$$

where $\pi_c^r$ represents the ratio of the number of samples in class $c$ to the total number of samples across all classes, $\sigma$ denotes the sigmoid activation function, $\alpha$ and $\beta$ are hyper-parameters for re-adjusting the scaling weight. Assume $w = [w_1, w_2, ..., w_{c_k+c_n}]$ is a vector of length $|c_k + c_n|$, the scaled logit $\hat{f}(x_j^u)$ for an unlabeled sample $x^u$ from known or novel class can be given as:

$$\hat{f}(x_j^u) = w \cdot f(x_j^u) \tag{5}$$

Due to the uniform threshold applied to pseudo-label masking Cai et al. (2022); Zheng et al. (2022), we propose leveraging an estimation of the distribution of unlabeled data to scale the logits. This method facilitates the adjustment of the masking level for samples across different predicted categories. More specifically, it limits the participation in loss calculation of samples from categories with a higher number of predicted instances, while increasing the participation rate of samples from categories with less estimated amounts. This adjustment aids the model in focusing more on learning from samples that are biased towards the tail classes Ma et al. (2024).

## 3.4 OVERALL OPTIMIZATION OBJECTIVE

Inspired by Wei & Gan (2023b), the logits of original pseudo-label $q(x_j^u)$ corresponding to known classes in the part used for generating refined pseudo-labels $\widetilde{q}(x_j^u)$ are adjusted:

$$\widetilde{q}(x_j^u) = \arg\max\left(f(x_j^u)_{[1,c_k]} - \tau_2 \cdot \log \Omega_{q(x_j^u)}\right), \tau_2 > 0 \tag{6}$$

where $f(x_j^u)_{[1,c_k]}$ represents the operation of adjusting the logits for the known categories in the generated pseudo-labels, in the same manner as is done with labeled data. Therefore, for the loss calculation in adjusted branches of labeled and unlabeled data based on cross-entropy loss Ren et al. (2020); Sohn et al. (2020) the loss function of adjusted branch can be defined as follows:

$$\mathcal{L}_{b\_ce} = -\sum_{i=1}^{m_k^l} \log\left(\frac{e^{f_y(x_i^l) + \tau \log \Omega_{y_i^l}}}{\sum_{c=1}^{c_k} e^{f_c(x_i^l) + \tau \log \Omega_c}}\right) + \sum_{j=1}^{m_k^u + m_n^u} \mathbb{M}(x_j^u)\mathcal{L}_{ce}\left(\hat{f}(x_j^u), \widetilde{q}(x_j^u)\right) \tag{7}$$

where $\mathcal{L}_{ce}$ represents standard Cross Entropy loss and $\mathbb{M}(x_j^u) := \mathbb{I}\left(\max(\delta(\hat{f}(x_j^u)) \geq \rho)\right)$ is the sample masks which selects unlabeled samples with predicted confidence levels exceeding a predefined threshold $\rho$. In detail, $\delta(\cdot)$ and $\mathbb{I}(\cdot)$ denote Softmax function and indicator function Wei & Gan (2023b). Therefore, there are two types of losses in neural networks that need to be optimized. The first is the original Cross Entropy loss calculation for both labeled and unlabeled data; The second is the balanced Cross Entropy loss calculation after logit adjustment, which can be given as follows:

$$\mathcal{L}_{rolssl} = \mathcal{L}_{pair} + \lambda_1 \mathcal{L}_{ce} + \lambda_2 \mathcal{L}_{b\_ce} + \mathcal{L}_{reg} \tag{8}$$

where $\lambda_1$ and $\lambda_2$ are trade-off parameters, and they are generally set to $\lambda_1 = \lambda_2 = 0.5$ in order to keep the scale of the loss consistent with OLSSL design (detailed in Ablation 4.4 and Appendix C).

## 4 EXPERIMENTS

### 4.1 IMPLEMENTATIONS

**Datasets**: To evaluate the effectiveness of OpenLDN, we conduct experiments on five widely-used benchmark datasets: CIFAR-10 Krizhevsky et al. (2010a), SVHN Netzer et al. (2011), CIFAR-100 Krizhevsky et al. (2010b), ImageNet-100 Deng et al. (2009), Tiny ImageNet Le & Yang (2015), and the Oxford-IIIT Pet dataset Parkhi et al. (2012). The CIFAR-10 and CIFAR-100 datasets each contain 60,000 images (split into 50,000 for training and 10,000 for testing), with 10 and 100 categories, respectively. SVHN dataset contains 73257 digits for training, 26032 digits for testing, with 10 classes. The ImageNet-100 dataset consists of 100 categories selected from ImageNet. Tiny ImageNet includes 100,000 training images and 10,000 validation images across 200 classes. The Oxford-IIIT Pet dataset comprises images from 37 categories, divided into 3,718 training and 3,707 testing images.

**Implementation Details**: We employ ResNet-18 as our primary feature extractor across all experiments except in instances involving ImageNet-100, where ResNet-50 is utilized. Our pairwise similarity prediction network, utilizing an MLP with a single 100-dimensional hidden layer, and a linear classifier, forms the basis of our feature extraction architecture. We train the network to discover novel classes over 50 epochs with batch sizes of 200 and 480 for ImageNet-100. For CIFAR-10 dataset, SGD optimizer is employed and the Adam optimizer is used consistently throughout the training process for the remaining five datasets. The learning rates are set at 5e-4 for the feature extractor and 1e-2 for ImageNet-100. In order to boost performance, we incorporate Mixmatch, a well-regarded closed-world SSL methods, during the second stage of training to enhance data balance and pseudo-label accuracy for each class during iterative self-labeling sessions. More details on these implementation strategies and parameter settings, e.g. $N_c$, $\tau_1$, can be found in Appendix A.

**Evaluation Metrics**: We assess accuracy for known classes using standard measures. For novel classes, we evaluate clustering accuracy and employ the Hungarian algorithm for accurate prediction alignment and ground truth labels matching before final accuracy calculations. The effectiveness of the proposed method is further demonstrated by joint accuracy measurements on both known and novel classes utilizing the Hungarian algorithm and normalized mutual information (**NMI**).

**OSSL Baselines:** We employ FixMatch Sohn et al. (2020), DS$^3$L Guo et al. (2020), CGDL Sun et al. (2020), DTC Han et al. (2019), RankStats Han et al. (2020), UNO Fini et al. (2021), ORCA Cao et al. (2022) and OpenLDN Rizve et al. (2022) to compare OSSL baselines with ROLSSL methods.

Table 2: Accuracy on the CIFAR-10, CIFAR-100, and ImageNet-100 datasets with 50% known and 50% novel classes under three different long-tailed conditions.

| | CIFAR-10 | | | CIFAR-100 | | | ImageNet100 | | |
|---|---|---|---|---|---|---|---|---|---|
| | Known | Novel | All | Known | Novel | All | Known | Novel | All |
| **Method** | *Semi-supervised & Open-world* | | | | | | | | |
| FixMatch (NIPS'20) | 71.5 | 50.4 | 49.5 | 39.6 | 23.5 | 20.3 | 65.8 | 36.7 | 34.9 |
| DS$^3$L (PMLR'20) | 77.6 | 45.3 | 40.2 | 55.1 | 23.7 | 24.0 | 71.2 | 32.5 | 30.8 |
| CGDL (CVPR'20) | 72.3 | 44.6 | 39.7 | 49.3 | 22.5 | 23.5 | 67.3 | 33.8 | 31.9 |
| DTC (CVPR'19) | 53.9 | 39.5 | 38.3 | 31.3 | 22.9 | 18.3 | 25.6 | 20.8 | 21.3 |
| RankStats (ICLR'20) | 86.6 | 81.0 | 82.9 | 36.4 | 28.4 | 23.1 | 47.3 | 28.7 | 40.3 |
| UNO (ICCV'21) | 91.6 | 69.3 | 80.5 | 68.3 | 36.5 | 51.5 | — | — | — |
| ORCA (ICLR'22) | 88.2 | 90.4 | 89.7 | 66.9 | 43.0 | 48.1 | 89.1 | 72.1 | 77.8 |
| OpenLDN (ECCV'22) | 95.2 | 92.7 | 94.0 | 73.3 | 46.8 | 60.1 | — | — | — |
| **Method** | ***Long-tailed (Consistent)** & Semi-supervised & Open-world* | | | | | | | | |
| OpenLDN | 44.2 | 12.7 | 28.4 | 31.3 | 11.6 | 22.9 | **18.7** | 4.2 | 12.5 |
| **Ours** | **46.7** | **38.7** | **46.2** | **32.0** | **18.9** | **25.4** | 17.1 | **8.1** | **14.0** |
| NMI (OpenLDN) | - | 0.196 | 0.224 | - | 0.391 | 0.389 | - | 0.256 | 0.281 |
| **NMI (Ours)** | - | **0.564** | **0.464** | - | **0.427** | **0.424** | - | **0.285** | **0.305** |
| **Method** | ***Long-tailed (Reversed)** & Semi-supervised & Open-world* | | | | | | | | |
| OpenLDN | 48.5 | 1.2 | 26.6 | 27.2 | 19.7 | 24.0 | **18.8** | 4.8 | 13.7 |
| **Ours** | **49.8** | **38.3** | **44.1** | **31.8** | **20.7** | **26.8** | 13.9 | **13.6** | **15.2** |
| NMI (OpenLDN) | - | 0.068 | 0.125 | - | 0.378 | 0.372 | - | 0.256 | 0.287 |
| **NMI (Ours)** | - | **0.394** | **0.405** | - | **0.457** | **0.438** | - | **0.302** | **0.323** |
| **Method** | ***Long-tailed (Uniform)** & Semi-supervised & Open-world* | | | | | | | | |
| OpenLDN | 44.5 | 3.8 | 24.2 | 21.4 | 7.6 | 14.8 | **13.6** | 3.3 | 10.4 |
| **Ours** | **47.1** | **53.9** | **50.5** | **22.8** | **8.2** | **15.9** | 11.2 | **7.3** | **11.4** |
| NMI (OpenLDN) | - | 0.111 | 0.155 | - | 0.289 | 0.280 | - | 0.238 | 0.252 |
| **NMI (Ours)** | - | **0.429** | **0.399** | - | **0.351** | **0.323** | - | **0.271** | **0.273** |

## 4.2 DISCUSSIONS ON EXPERIMENTAL RESULTS

In Tables 2 and 3, we compare the performance of OpenLDN and the proposed method across six experimental benchmark datasets under Long-tailed (*Consistent*), Long-tailed (*Reversed*), and Long-tailed (*Uniform*) conditions. The results demonstrate that the proposed method consistently outperforms OpenLDN across almost all datasets in terms of known class, novel class, and overall class recognition accuracy. Furthermore, the proposed method is able to achieve a more stable training process (detailed in Section 4.3). Specifically, under the Long-tailed (*Consistent*) condition, the proposed method shows significant improvements in CIFAR10, CIFAR100, SVHN, and Oxford-IIIT Pet datasets. Under the Long-tailed (Reversed) condition, the proposed method exhibits substantial enhancements in recognizing novel classes across all datasets, with particularly notable performance in CIFAR10, ImageNet100, and SVHN. Under the Long-tailed (*Uniform*) condition, the proposed method significantly surpasses OpenLDN in CIFAR10 and SVHN datasets. Moreover, the proposed method achieves higher normalized mutual information (NMI) scores in the recognition of both novel classes and overall samples, which further attests to the superior performance of the model presented in this paper. These results indicate that the proposed method not only excels in recognizing known classes but also demonstrates exceptional performance in novel and overall class recognition, highlighting its robustness and adaptability in handling complex, long-tailed distributions. Overall, the proposed method showcases superior accuracy and broad applicability in ROLSSL tasks, proving its effectiveness in addressing the challenges posed by diverse datasets and varying class distributions. The proposed method provides a potentially viable solution within the ROLSSL framework.

## 4.3 DISCUSSION ABOUT OSSL METHOD IN ROLSSL SETTINGS

In the previous section, we mentioned that directly applying the OSSL scheme within the ROLSSL framework leads to a decline in the recognition ability for novel classes as the training progresses.

Table 3: Accuracy on the Tiny ImageNet, Oxford-IIIT Pet, and SVHN datasets with 50% known and 50% novel classes under three different long-tailed conditions.

| | Tiny ImageNet | | | Oxford-IIIT Pet | | | SVHN | | |
|---|---|---|---|---|---|---|---|---|---|
| | Known | Novel | All | Known | Novel | All | Known | Novel | All |
| **Method** | *Semi-supervised & Open-world* | | | | | | | | |
| DTC (CVPR'19) | 28.8 | 16.3 | 19.9 | 20.7 | 16.0 | 13.5 | 90.3 | 65.0 | 81.0 |
| RankStats (ICLR'20) | 5.7 | 5.4 | 3.4 | 12.6 | 11.9 | 11.1 | 96.3 | 96.1 | 96.2 |
| UNO (ICCV'21) | 46.5 | 15.7 | 30.3 | 49.8 | 22.7 | 34.9 | 85.4 | 74.3 | 79.0 |
| OpenLDN (ECCV'22) | 52.3 | 19.5 | 36.0 | 67.1 | 27.3 | 47.7 | 95.7 | 87.2 | 92.6 |
| **Method** | ***Long-tailed (Consistent)** & Semi-supervised & Open-world* | | | | | | | | |
| OpenLDN | 13.7 | 7.7 | 12.1 | 12.7 | 2.1 | 10.5 | 59.9 | 0.5 | 37.9 |
| **Ours** | **15.6** | **10.0** | **13.8** | **14.9** | **6.7** | **12.0** | **67.5** | **35.4** | **56.2** |
| NMI (OpenLDN) | - | 0.421 | 0.418 | - | 0.146 | 0.134 | - | 0.099 | 0.236 |
| **NMI (Ours)** | **-** | **0.445** | **0.441** | **-** | **0.157** | **0.147** | **-** | **0.292** | **0.470** |
| **Method** | ***Long-tailed (Reversed)** & Semi-supervised & Open-world* | | | | | | | | |
| OpenLDN | 13.8 | 9.5 | 13.0 | **13.2** | 1.8 | 10.0 | 31.4 | 15.3 | 25.9 |
| **Ours** | **17.3** | **10.3** | **14.9** | 13.0 | **8.5** | **12.5** | **77.1** | **20.2** | **48.6** |
| NMI (OpenLDN) | - | 0.410 | 0.422 | - | 0.127 | 0.119 | - | 0.064 | 0.121 |
| **NMI (Ours)** | **-** | **0.426** | **0.427** | **-** | **0.165** | **0.154** | **-** | **0.170** | **0.465** |
| **Method** | ***Long-tailed (Uniform)** & Semi-supervised & Open-world* | | | | | | | | |
| OpenLDN | 9.5 | 9.0 | 9.0 | **10.7** | 1.4 | 9.5 | 56.3 | 2.9 | 36.6 |
| **Ours** | 9.2 | **10.6** | **9.6** | 10.2 | **4.2** | **10.2** | **58.4** | **31.9** | **49.7** |
| NMI (OpenLDN) | - | 0.385 | 0.379 | - | **0.126** | **0.114** | - | 0.050 | 0.210 |
| **NMI (Ours)** | **-** | **0.401** | **0.390** | **-** | 0.123 | 0.113 | **-** | **0.297** | **0.451** |

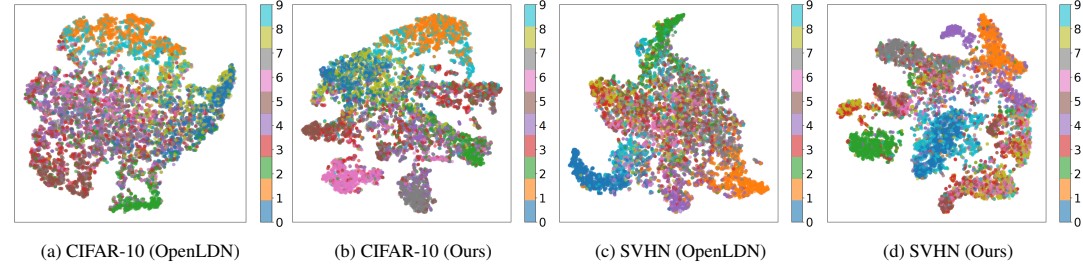

| (a) CIFAR-10 (OpenLDN) | (b) CIFAR-10 (Ours) | (c) SVHN (OpenLDN) | (d) SVHN (Ours) |
|---|---|---|---|

Figure 2: Figures (a) and (c) show the t-SNE visualizations of OpenLDN on the CIFAR-10 and SVHN datasets, respectively. Figures (b) and (d) present the t-SNE visualizations of the proposed method on the CIFAR-10 and SVHN datasets. It is evident that DPLA demonstrates better recognition performance compared to OpenLDN.

Here, we illustrate this phenomenon by examining the recognition accuracy of known, novel, and overall samples during the training process on the SVHN dataset under a consistent setting using the OLSSL scheme. As shown in the Figure 5 of Appendix B, the OLSSL scheme, OpenLDN, consistently exhibits low recognition performance for novel classes, dropping to zero recognition accuracy for novel classes at around the 16th epoch and failing to recover this ability throughout the subsequent training. In contrast, the dual-stage post-hoc logit adjustment (DPLA) proposed in this paper effectively addresses this issue. DPLA maintains the ability to recognize novel class samples and can achieve recognition accuracy close to or even exceeding that of OpenLDN for all samples at certain stages, demonstrating the effectiveness of DPLA. Moreover, DPLA consistently achieves higher recognition accuracy for both known classes and overall samples compared to OpenLDN, without experiencing a gradual decline in accuracy. This indicates that DPLA is well-suited for the ROLSSL framework and significantly improves accuracy. It is also noteworthy that OpenLDN rarely regains the ability to recognize novel class samples as training progresses; however, due to random seed variations, OpenLDN has a slight chance of achieving very low recognition accuracy for novel class samples in the final few epochs, thereby transitioning into a close-world training phase. To highlight the performance differences between OpenLDN and DPLA under their optimal conditions, we selected the best performance of OpenLDN when it had a favorable initialization and could transition into the close-world training phase for comparison.

## 4.4 ABLATION STUDY

**Method Design:** We utilize the SVHN dataset to investigate the impact of each design within each DPLA on model performance. We employ OpenLDN as the performance baseline for model comparisons. As observed, the inclusion of logit adjustment in the first stage significantly improves the performance for known, novel, and overall categories, with accuracy for the novel category increasing by up to 30%. The introduction of the second stage and pseudo-label adjustment further enhances model performance, though the improvement is less pronounced and shows a diminishing trend.

Table 4: Performance comparison of ablation experiments designed to explore the role of each stage of DPLA. Baseline is OpenLDN.

| Method | Known | Novel | All |
|---|---|---|---|
| Baseline | 59.9 | 0.5 | 37.9 |
| + First Stage | 65.1 | 32.5 | 54.4 |
| + Second Stage | 67.2 | 35.3 | 55.7 |
| +PLR (DPLA) | 67.5 | 35.4 | 56.2 |

**First-stage Scaling Factor:** The reason for designing the scaling factor $10 \cdot (\lceil \mathcal{C}/\mathcal{C}_{base} \rceil) \cdot \sqrt{\mathcal{S}/\mathcal{S}_{base}}$ in the first stage is that original post-hoc logit adjustment design only achieves expected performance in datasets with fewer categories, such as CIFAR-10 and SVHN. However, for datasets like CIFAR-100 and ImageNet-100, it suppresses model performance and fails to improve accuracy under data imbalance conditions. Therefore, we design the first-stage scaling factor based on sample frequency, total number of dataset categories, and data size. From Figures 3 and 4, it can be concluded that when PLA is applied directly to ROLSSL without any modifications, the model performance is even lower than the baseline performance obtained by directly applying OpenLDN to ROLSSL. As the scaling factor increases to the multiples set in this study, model accuracy gradually rises and eventually surpasses the baseline. However, when scaling factor continues to increase, model performance declines, demonstrating the rationality and effectiveness of our proposed method design.

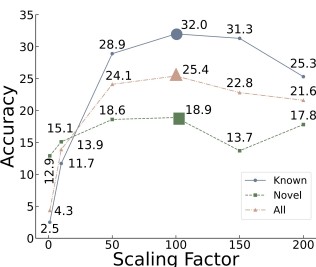

Figure 3: Ablation study on the performance of CIFAR-100 for the Scaling Factor.

Figure 4: Ablation study on the performance of ImageNet-100 for the Scaling Factor.

Table 5: Ablation study on the model performance for trade-off parameters $\lambda_1$ and $\lambda_2$.

| $\lambda_1, \lambda_2$ | Novel | All |
|---|---|---|
| (0.2, 0.8) | 25.7 | 38.5 |
| (0.3, 0.7) | 30.3 | 42.0 |
| (0.4, 0.6) | 33.8 | 44.2 |
| (0.5, 0.5) | **38.7** | **46.2** |
| (0.6, 0.4) | 36.3 | 45.8 |
| (0.7, 0.3) | 32.9 | 45.4 |
| (0.8, 0.2) | 29.4 | 41.3 |

**Trade-off Parameter:** The design of the final loss optimization objective involves setting $\lambda_1$ and $\lambda_2$. We conduct an investigation based on the CIFAR-10 dataset, and it is observed that for this dataset, $\lambda_1 = \lambda_2 = 0.5$ is the optimal setting. Under other settings, the model performance shows some degree of decline, and this performance pattern is consistent across most other datasets. In the experiments, $\lambda_1 + \lambda_2 = 1$ should be satisfied, mainly to maintain the numerical scale of each loss similar to the original design in $\mathcal{L}_{OSSL}$, which is beneficial for recognizing novel category data.

## 5 CONCLUSION

Realistic open-world long-tailed semi-supervised learning (ROLSSL) provides a more realistic experimental setup for open-world semi-supervised learning by considering various data imbalance relationships among known and novel categories, as well as the high cost of obtaining labeled data in real-world scenarios. Building on the traditional post-hoc logit adjustment, this paper proposes dual-stage post-hoc logit adjustment (DPLA). By integrating factors such as sample frequency and the total number of categories, this approach better utilizes both labeled and unlabeled data. The proposed method significantly improves model performance under the ROLSSL setting, outperforming other comparative approaches and providing a simple yet strong performance baseline for this task.

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

## A   APPENDIX ON EXPERIMENTAL SETTINGS

Due to computational constraints, we evaluated the performance of the base stage only on the ImageNet-100 and Oxford-IIIT Pets datasets. For all datasets that underwent closed-world stage training, the total number of training epochs is 256, with a batch size of 64 and a learning rate of 0.002. For the experiments for all of the datasets, the masking threshold is set to 0.5 uniformly. $\mathcal{C}_{base}$ is set to 10 and $\mathcal{S}_{base}$ is set to $32 \times 32$ which conforms to the resolution of single images of CIFAR-10. $\mathcal{C}$ and $\mathcal{S}$ are corresponding parameters of the estimated dataset.

**CIFAR-10 dataset:** In the context of the CIFAR-10 study, our methodology is evaluated using the setting: $N_1 = 500, H_1 = 4000, M_1 = 4500$. We establish the three kinds of imbalance ratios at $\gamma_k^l = \gamma_k^u = \gamma_n^u = 100$. Moreover, maintaining a constant $\gamma_k^l = \gamma_k^u = 100$, we further explore our approach under varying conditions $\gamma_n^u = 1/100$ and $M_1 = M_2 = ... = M_{c_n} = 1500$, to simulate both reversed and uniform distributions of unlabeled novel-class data classes. Besides, for the remaining experimental parameters $\lambda_1 = 0.5, \lambda_2 = 0.5, \tau_1 = 2, \tau_2 = 2, \alpha = 1.2$ and $\beta = 0.8$.

**CIFAR-100 dataset:** In the context of the CIFAR-100 study, our methodology is evaluated using the setting: $N_1 = 50, H_1 = 400, M_1 = 450$. We establish the three kinds of imbalance ratios at $\gamma_k^l = \gamma_k^u = \gamma_n^u = 100$. Moreover, maintaining a constant $\gamma_k^l = \gamma_k^u = 100$, we further explore our approach under varying conditions $\gamma_n^u = 1/100$ and $M_1 = M_2 = ... = M_{c_n} = 150$, to simulate both reversed and uniform distributions of unlabeled novel-class data classes. Besides, for the remaining experimental parameters $\lambda_1 = 0.5, \lambda_2 = 0.5, \tau_1 = 1, \tau_2 = 1, \alpha = 1.05$ and $\beta = 0.95$.

**ImageNet-100 dataset:** In the context of the ImageNet-100 study, our methodology is evaluated using the setting: $N_1 = 75, H_1 = 600, M_1 = 675$. We establish the three kinds of imbalance ratios at $\gamma_k^l = \gamma_k^u = \gamma_n^u = 100$. Moreover, maintaining a constant $\gamma_k^l = \gamma_k^u = 100$, we further explore our approach under varying conditions $\gamma_n^u = 1/100$ and $M_1 = M_2 = ... = M_{c_n} = 225$, to simulate both reversed and uniform distributions of unlabeled novel-class data classes. Besides, for the remaining experimental parameters $\lambda_1 = 0.8, \lambda_2 = 0.2, \tau_1 = 1, \tau_2 = 1, \alpha = 1.05$ and $\beta = 0.95$.

**Tiny ImageNet dataset:** In the context of the Tiny ImageNet study, our methodology is evaluated using the setting: $N_1 = 50, H_1 = 400, M_1 = 450$. We establish the three kinds of imbalance ratios at $\gamma_k^l = \gamma_k^u = \gamma_n^u = 10$. Moreover, maintaining a constant $\gamma_k^l = \gamma_k^u = 100$, we further explore our approach under varying conditions $\gamma_n^u = 1/100$ and $M_1 = M_2 = ... = M_{c_n} = 150$, to simulate both reversed and uniform distributions of unlabeled novel-class data classes. Besides, for the remaining experimental parameters $\lambda_1 = 0.5, \lambda_2 = 0.5, \tau_1 = 1, \tau_2 = 1, \alpha = 1.2$ and $\beta = 0.8$.

**Oxford-IIIT Pet dataset:** In the context of the Oxford-IIIT Pet study, our methodology is evaluated using the setting: $N_1 = 20, H_1 = 60, M_1 = 80$. We establish the three kinds of imbalance ratios at $\gamma_k^l = \gamma_k^u = \gamma_n^u = 10$. Moreover, maintaining a constant $\gamma_k^l = \gamma_k^u = 100$, we further explore our approach under varying conditions $\gamma_n^u = 1/100$ and $M_1 = M_2 = ... = M_{c_n} = 20$, to simulate both reversed and uniform distributions of unlabeled novel-class data classes. Besides, for the remaining experimental parameters $\lambda_1 = 0.5, \lambda_2 = 0.5, \tau_1 = 1, \tau_2 = 1, \alpha = 1.05$ and $\beta = 0.95$.

**SVHN dataset:** In the context of the SVHN study, our methodology is evaluated using the setting: $N_1 = 500, H_1 = 4000, M_1 = 4500$. We establish the three kinds of imbalance ratios at $\gamma_k^l = \gamma_k^u = \gamma_n^u = 100$. Moreover, maintaining a constant $\gamma_k^l = \gamma_k^u = 100$, we further explore our approach under varying conditions $\gamma_n^u = 1/100$ and $M_1 = M_2 = ... = M_{c_n} = 1500$, to simulate both reversed and uniform distributions of unlabeled novel-class data classes. Besides, for the remaining experimental parameters $\lambda_1 = 0.5, \lambda_2 = 0.5, \tau_1 = 2, \tau_2 = 2, \alpha = 1.2$ and $\beta = 0.8$.

## B   OSSL METHOD PERFORMANCE IN ROLSSL SETTINGS.

To better observe the impact of the ROLSSL setting on previous OLSSL methods, we visualize the classification performance of the OpenLDN and our proposed DPLA methods on the SVHN dataset as epochs increase. In this Figure 5, circles and triangles represent DPLA and OpenLDN, respectively, while blue, yellow, and green represent the accuracy of known classes, novel classes, and all classes,

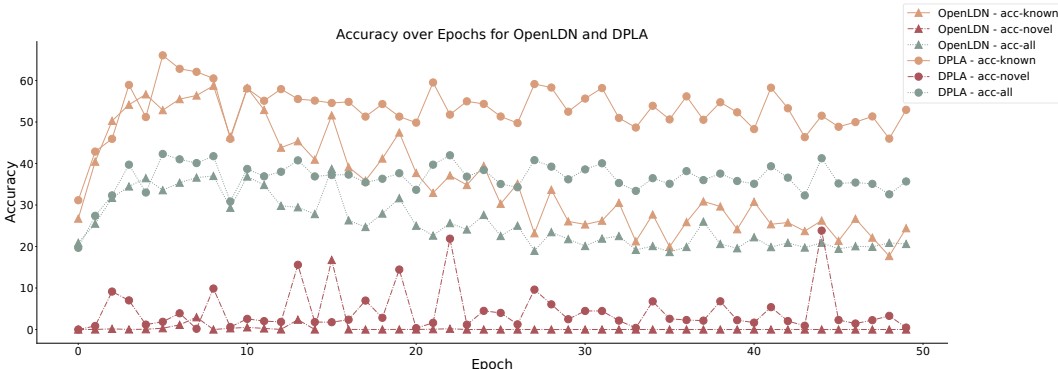

Figure 5: OSSL method performance in ROLSSL settings.

respectively. It can be seen that for known classes, the accuracy of OpenLDN gradually decreases as epochs increase, whereas the accuracy of our proposed DPLA steadily rises. For the prediction of novel classes, OpenLDN's performance approaches zero, while DPLA significantly outperforms OpenLDN, although it exhibits considerable fluctuations. Overall, OpenLDN performs very poorly under our proposed ROLSSL setting, especially in recognizing novel classes, while our proposed DPLA method far surpasses OpenLDN and effectively recognizes novel classes.

## C  APPENDIX ON PSEUDO-ALGORITHM FOR ROLSSL

Here we provide pseudo-code for the method proposed in this paper to clarify the steps of it.

---

**Algorithm 1** Dual-Stage Post-hoc Logit Adjustment for ROLSSL

---

**Require:** $\mathcal{D}_k^l, \mathcal{D}_k^u, \mathcal{D}_n^u, \alpha, \beta, \tau_1, \tau_2, \lambda_1, \lambda_2$
**Ensure:** Adjusted Model Logits
  1: **Initialize:** Set hyper-parameters $\alpha, \beta, \tau_1, \tau_2, \lambda_1, \lambda_2$
  2: Define datasets $\mathcal{D}_k^l$ (labeled), $\mathcal{D}_k^u, \mathcal{D}_n^u$ (unlabeled)
  3: **First-stage Logit Adjustment:**
  4: **for** $x^l$ in $\mathcal{D}_k^l$ **do**
  5:      Compute logit factor $\Omega$ from sample frequency and class count
  6:      Adjust logits: logit $\leftarrow f(x^l) - \tau_1 \log(\Omega)$
  7: **end for**
  8: **Second-stage Logit Adjustment:**
  9: **for** $x^u$ in $\mathcal{D}_k^u \cup \mathcal{D}_n^u$ **do**
10:      Estimate class distribution and adjust logits with $\alpha, \beta$:

$$\hat{f}(x^u) \leftarrow w \cdot f(x^u)$$

11: **end for**
12: **Pseudo-Label Generation:**
13: **for** $x^u$ in $\mathcal{D}_k^u \cup \mathcal{D}_n^u$ **do**
14:      Generate pseudo-labels and mask dominant classes
15: **end for**
16: **Training:**
17: Train model using cross-entropy loss and $\tau_2$-adjusted pseudo-labels
18: Optimize loss:
$$\text{Loss} \leftarrow \mathcal{L}_{\text{pair}} + \lambda_1 \mathcal{L}_{\text{ce}} + \lambda_2 \mathcal{L}_{\text{b\_ce}} + \mathcal{L}_{\text{reg}}$$
19: **Evaluation:** Assess performance using accuracy and mutual information

---

