# OpenReview forum: "Towards Realistic Long-tailed Semi-supervised Learning in an Open World"
_ICLR.cc/2025/Conference — ICLR 2025 Conference Withdrawn Submission_

### Official Review · Reviewer_ZhRQ · 2024-10-28

**Soundness:** 2
**Presentation:** 2
**Contribution:** 2
**Rating:** 5
**Confidence:** 4

**Summary:**

This paper aims to solve a complex task in semi-supervised learning, i.e., the long-tailed label distribution and the presence of novel classes in unlabeled data. The paper presents a simple solution for this task based on techniques such as logit adjustment, and entropy maximization.

**Strengths:**

1. The studied problem in this paper is practical and underexplored. Most previous studies focus on long-tailed or open-world semi-supervised learning, without considering both.
2. The proposed approach is simple and easy to implement. This paper improves previous methods by proposing a dual-stage post-hoc logit adjustment technique, which does not involve complex strategies or additional learnable parameters.
3. Experiments on six datasets show the advantage of the proposed approach against previous methods.

**Weaknesses:**

1. The novelty of the proposed method seems limited. As stated in the *Strengths*, the proposed method is simple and easy to implement because it is pretty much built upon existing techniques. However, this also makes the contribution of the method minor.

2. The rationale behind the method is not well-explained. It is unclear why the logit adjustment strategies for labeled and unlabeled data are different. Also, the design of Eq. (3) is not straightforward and needs in-depth understanding and justification. In Eq. (4), the impact of hyperparameters $\alpha$ and $\beta$ are not studied.

3. The writing needs improvement. For instance, the paper should include the definition of $\mathcal{L}\_{pair}$ and $\mathcal{L}\_{reg}$ to be self-contained.

4. It would be better if the paper could include real-world semi-supervised datasets which follow long-tailed label distribution and also include novel classes in unlabeled data.

5. It seems that the logit adjustment strategy in open-world semi-supervised learning has been considered in [1].

[1] Bridging the Gap: Learning Pace Synchronization for Open-World Semi-Supervised Learning. IJCAI 2024.

**Questions:**

1. The analysis of hyperparameters $\tau_1$ and $\tau_2$ are missing.

---

### Official Review · Reviewer_v2SP · 2024-11-03

**Soundness:** 3
**Presentation:** 2
**Contribution:** 2
**Rating:** 3
**Confidence:** 4

**Summary:**

This paper mainly focuses on the more challenging SSL setting - open-world long-tailed SSL (OLSSL). The authors propose a more realistic setting than the existing OLSSL: labeled samples are much less than the unlabeled ones. Based on the existing OLSSL method, this work proposes a dual-stage post-hoc logit adjustment method to calibrate and optimize the model. Several experiments are conducted for verification.

**Strengths:**

SSL in a more realistic setting is proposed and explored in this work.

**Weaknesses:**

1. Since readers may not be quite familiar with the development of open-world long-tail SSL, it is suggested that more background about the existing OLSSL algorithms be included in Section 3.2, especially for the definition of Eq(1).
2. The dual-stage post-hoc logit adjustment section is not easy to read. Specifically,
- why the aremax is used in Eq(2)? According to the Fig 1, it is expected to be a distribution adjustment.
- What is the definition of F_{y_i^l} in Eq(3)? If the labeled samples are limited, will this sample frequency be biased? How C_base and S_base are defined?
- What is the relationship of the equation in L244 and the Eq(2)?
- for unlabeled samples, how the sample ratio \pi_c^r in Eq(4) is obtained when the class of the unlabeled samples is unknown? How alpha and beta are defined? What is the meaning of (alpha-beta) in Eq(4)?
3. Does the \tau in Eq(7) denote the ones in Eq(2)? If so, why the operation become plus but not minus as in Eq(2)?
4. The proposed method is argued to be a two-stage method, does it mean training labeled and unlabeled samples in different stages? However, according to Eq(8), the model seems optimized on labeled and unlabeled samples together.
5. In Figure 1, the right side of the diagram shows a question mark ("?") symbol, does it represent the unlabeled samples? If so, why does the labeled branch also include a depiction of unlabeled samples? Is there a particular significance or reason for this choice in the illustration?
6. The paper seems primarily focused on addressing the long-tailed problem while offering little innovation for the open-world problem.
7. In the defined Realistic Open-World Long-Tailed Semi-Supervised Learning (ROLSSL) framework, it is assumed that the number of novel classes is known (L200). The proposed method leverages this prior knowledge for model calibration (Eq(3)). However, this assumption seems somewhat unrealistic, as it is uncommon to have prior knowledge of the exact number of novel classes present in the unlabeled samples.

**Questions:**

This work may need more effort to refine its writing and the methodology sections need more explanations for equations as mentioned above.

---

### Official Review · Reviewer_YL7M · 2024-11-04

**Soundness:** 3
**Presentation:** 3
**Contribution:** 2
**Rating:** 5
**Confidence:** 4

**Summary:**

The paper introduces a approach called Dual-stage Post-hoc Logit Adjustments (DPLA) to tackle the challenges in Realistic Open-world Long-tailed Semi-supervised Learning (ROLSSL). DPLA is a two-stage logit adjustment technique designed to improve model performance on both frequent and infrequent (tail) classes.  In the first stage, logits are adjusted based on the sample frequency and class count, which helps balance predictions across known classes. In the second stage, adjustments are dynamically refined using the predicted class distribution from unlabeled data, with a particular focus on emphasizing tail classes to counter bias towards dominant classes. This dual-stage approach enables better recognition in long-tailed, semi-supervised, and open-world settings, especially where labeled data is sparse and imbalanced. Key experimental results highlight that the DPLA approach consistently surpasses the OpenLDN baseline in both known and novel class recognition, achieving up to 50.1% improvement experiments on datasets such as CIFAR100 and ImageNet100. They also show the robustness and adaptability of the proposed method across different data imbalance conditions.

**Strengths:**

1. The problem addressed by this paper is highly practical, as determining the distributional relationships between known and novel categories is indeed nearly impossible in real-world scenarios.

2. The paper is well-written and easy to follow.

3. Key experimental results demonstrate that the DPLA approach consistently outperforms the OpenLDN baseline in both known and novel class recognition, achieving up to a 50.1% improvement on datasets like CIFAR-100 and ImageNet-100. Authors further showcase the robustness and adaptability of the proposed method across varying levels of data imbalance.

**Weaknesses:**

1. A key concern is the complexity of the proposed pipeline, which involves multiple stages and numerous hyper-parameters. With additional stages and hyper-parameters to tune, it becomes more challenging to generalize this pipeline to new problem settings and datasets. For instance, as shown in Tables 3 and 4, using non-optimal scaling factors can significantly impact performance on CIFAR-100 and ImageNet-100, despite both datasets having the same number of classes. Have you explored methods to automatically tune the hyperparameters or analyzed which parameters are most sensitive?

2. Technical contributions. A primary contribution of this paper is the proposed logit adjustment method for pseudo-labels, which often exhibit class imbalance. However, the issue of naturally imbalanced pseudo-labels has been previously addressed by DebiasPL [1], which also applies a logit adjustment based technique to mitigate class imbalance in pseudo-labels, resulting in improved classification performance on long-tailed datasets. A discussion on the technical distinctions between this approach and DebiasPL would be beneficial.

3. While this paper presents experiments on several benchmarks to demonstrate performance gains, most datasets used are small to medium-scale, such as ImageNet-100 or CIFAR. I believe it is essential to include experiments on larger datasets, such as iNaturalist or ImageNet-1k or ImageNet-22k, to better assess the generalizability of the proposed method. It will be nice if you could provide any potential challenges or modifications needed to apply your method to larger datasets.

4. In addition to presenting results on novel and known subsets, I believe it is essential to include performance metrics for many-shot, medium-shot, and few-shot subsets. This would help clarify whether the current logit adjustment methods compromise performance on many-shot classes to improve results on few-shot classes—a common issue with most logit adjustment techniques. It would be valuable for the authors to discuss whether their proposed method faces this challenge and, if not, to explain why it avoids this problem. If it does, providing insights into potential strategies to address this limitation in future work would be beneficial.

5. I may suggest authors to include a specific breakdown of performance metrics for these subsets in their results tables. Add a discussion section that explicitly addresses the trade-offs between performance on different shot categories and potential strategies to balance these trade-offs. In my opinion, while improving performance on few-shot classes is important, it is equally crucial to avoid jeopardizing performance on many-shot classes, as they are more prevalent in real-world applications.

[1] Wang, Xudong, Zhirong Wu, Long Lian, and Stella X. Yu. "Debiased learning from naturally imbalanced pseudo-labels." In Proceedings of the IEEE/CVF Conference on Computer Vision and Pattern Recognition, pp. 14647-14657. 2022.

**Questions:**

Have you considered running experiments on iNaturalist? It has more classes than any of the benchmarks used in this paper and presents a more challenging fine-grained, long-tailed distribution.

---

### Official Review · Reviewer_3kwE · 2024-11-05

**Soundness:** 2
**Presentation:** 2
**Contribution:** 2
**Rating:** 3
**Confidence:** 5

**Summary:**

This paper studies a realistic open-world setting, namely open-world long-tailed semi-supervised learning (ROLSSL), where the distribution relationships between known and novel categories might be different. To solve this problem, the paper proposes dual-stage post-hoc logit adjustments. Specifically, it estimates the distributions of the unlabeled data and utilizes the estimations to re-adjust the logits. Comprehensive experiments validate the superiority of the method.

**Strengths:**

1. This paper is well-motivated and easy to follow.
2. The proposed problem setting is realistic and reflects the long-tailed nature of the real world.
3. This paper conducts comprehensive experiments to show the superiority of the method.

**Weaknesses:**

1. The comparative methods are outdated, and the most recent comparative method of this paper is ECCV 2022. In the literature, OSSL is also referred to as generalized category discovery (GCD). Some recent GCD methods, including GCD [R1] and SimGCD [R2], should be included in Table 2 and 3. There are also some works [R3, R4] in GCD considering long-tailed scenarios, which are highly related to this work and should be compared.
2. The novelty of this paper is limited. The basic loss functions $L_{pair}$ and $L_{reg}$ are well-known techniques in novel class discovery and GCD, while the logit adjustments are also prevalent methods in long-tailed classification. Simply applying model predictions in logit adjustments is naïve and might not provide great insights to the community.
3. The method still requires strong priors of the ground truth new class numbers, which is unrealistic.
4. Experiments with the ViT backbone should be conducted.
5. The notations should be concise and clear.

References:
[R1]. Generalized Category Discovery. CVPR 2022.
[R2]. Parametric Classification for Generalized Category Discovery: A Baseline Study. ICCV 2023.
[R3]. Novel Class Discovery for Long-tailed Recognition. TMLR 2023.
[R4]. Generalized Categories Discovery for Long-tailed Recognition. ICCV 2023, workshop.

**Questions:**

1. How to handle the scenarios where the number of new classes is unknow? It seems that the proposed method requires the prior class number, which might not realistic.
2. Could the authors summarize the main novelty intuitively?

---

### Official Review · Reviewer_vGbF · 2024-11-09

**Soundness:** 2
**Presentation:** 2
**Contribution:** 2
**Rating:** 3
**Confidence:** 3

**Summary:**

The paper addresses the limitations of existing open-world long-tailed semi-supervised learning (OLSSL) algorithms, which assume identical distributions between known and novel categories. It introduces a more realistic open-world long-tailed semi-supervised learning (ROLSSL) setting, where no assumption is made about the distribution relationship between known and novel classes. The authors propose a novel approach called dual-stage post-hoc logit adjustments, which dynamically adjusts predictive probabilities to mitigate category bias during learning. The method takes into account the frequency of samples, the number of categories, and the size of the data, improving performance by more selectively utilizing imbalanced unlabeled data. Experiments on CIFAR100 and ImageNet100 show some improvements in performance, demonstrating the effectiveness of the proposed method and setting a strong baseline for future research.

**Strengths:**

1. The combination of long-tailed, semi-supervised, and open-world learning is interesting.

**Weaknesses:**

1. The writing of this paper is not very clear. For example, 'S/N Consistency' in Tab. 1, a key feature of the proposed setting, is never explained.
2. Open-world Long-tailed Semi-supervised Learning seems quite similar to [*1], which is not discussed in related works. Therefore, the novelty of this paper may be weak.
3. Experiments may be insufficient. Only some tiny datasets are considered. Large-scale benchmarks (e.g., ImageNet-LT or iNaturalist) should be included.


[*1] SimPro: A Simple Probabilistic Framework Towards Realistic Long-Tailed Semi-Supervised Learning. ICML'24

**Questions:**

See Weaknesses.

---

### Note · Authors · 2024-11-12

I have read and agree with the venue's withdrawal policy on behalf of myself and my co-authors.